# Field Emission from Carbon Nanotubes on Titanium Nitride-Coated Planar and 3D-Printed Substrates

**DOI:** 10.3390/nano14090781

**Published:** 2024-04-30

**Authors:** Stefanie Haugg, Luis-Felipe Mochalski, Carina Hedrich, Isabel González Díaz-Palacio, Kristian Deneke, Robert Zierold, Robert H. Blick

**Affiliations:** 1Center for Hybrid Nanostructures (CHyN), Universität Hamburg, 22761 Hamburg, Germany; luis-felipe.mochalski@desy.de (L.-F.M.); chedrich@physnet.uni-hamburg.de (C.H.); igonzale@physnet.uni-hamburg.de (I.G.D.-P.); kdeneke@physnet.uni-hamburg.de (K.D.); rzierold@physik.uni-hamburg.de (R.Z.); rblick@physnet.uni-hamburg.de (R.H.B.); 2Deutsches Elektronen-Synchrotron (DESY), 22607 Hamburg, Germany

**Keywords:** field emission, carbon nanotubes, titanium nitride, atomic layer deposition, 3D-printed microstructures, direct laser writing, two-photon polymerization

## Abstract

Carbon nanotubes (CNTs) are well known for their outstanding field emission (FE) performance, facilitated by their unique combination of electrical, mechanical, and thermal properties. However, if the substrate of choice is a poor conductor, the electron supply towards the CNTs can be limited, restricting the FE current. Furthermore, ineffective heat dissipation can lead to emitter–substrate bond degradation, shortening the field emitters’ lifetime. Herein, temperature-stable titanium nitride (TiN) was deposited by plasma-enhanced atomic layer deposition (PEALD) on different substrate types prior to the CNT growth. A turn-on field reduction of up to 59% was found for the emitters that were generated on TiN-coated bulk substrates instead of on pristine ones. This observation was attributed exclusively to the TiN layer as no significant change in the emitter morphology could be identified. The fabrication route and, consequently, improved FE properties were transferred from bulk substrates to free-standing, electrically insulating nanomembranes. Moreover, 3D-printed, polymeric microstructures were overcoated by atomic layer deposition (ALD) employing its high conformality. The results of our approach by combining ALD with CNT growth could assist the future fabrication of highly efficient field emitters on 3D scaffold structures regardless of the substrate material.

## 1. Introduction

Carbon nanotubes (CNTs) are nowadays integrated as core pieces in a variety of applications, such as chemical sensors [1], reinforced polymeric materials [2], batteries [3], solar cells [4], and corrosion protective coatings [5], to name a few. Furthermore, CNT-based devices are a promising candidate for the development of commercial field emission (FE) electron sources facilitated by their distinctive structural appearance. Typically, their very large aspect ratio leads to a huge geometrical field enhancement factor, which causes an extremely high local electric field on the surface of the CNTs, easing the emission of electrons by FE [6]. Moreover, CNTs possess exceptional physical properties, i.e., a high intrinsic electrical conductivity of about 10^3^–10^5^ S/m [7], a thermal conductivity comparable to that of diamond [8], and a tensile strength that is nearly 100 times larger than that of steel [2]. One key factor that can limit the FE performance of the CNTs and, consequently, prevent their integration into commercial FE-based devices, is the substrate on which they are anchored. For the sake of convenience, the electrical contact to a CNT layer is typically established via the substrate in FE experiments. Therefore, a metal substrate would be the ideal choice to provide a low contact resistance resulting in an efficient electron supply towards the CNTs. Moreover, a metallic substrate would allow for adequate heat dissipation, preventing emitter–substrate bond degradation [9,10]. However, the chemical vapor deposition (CVD) growth of CNTs on metal substrates is known to be challenging in contrast to substrates like silicon or quartz. Metals have lower melting points that can be incompatible with CVD growth temperatures, they show an enhanced probability for chemical or physical reaction with the catalyst material (such as iron, nickel, or cobalt) needed for CNT growth, and their higher surface roughness provokes inhomogeneous catalyst particle distributions [10,11]. These substrate properties can have a strong influence on the resulting CNT structure and yield, or even prevent CNT growth on the metal substrate.

In previous studies, it was found that titanium nitride (TiN) is a suitable substrate material for the CVD growth of CNT field emitters because it is a stable ceramic material that is known for its high melting point (~3200 °C), low electrical resistivity (<1 mΩ∙cm for sputtered TiN films), extreme hardness, and its good corrosion and diffusion resistance [12,13,14,15,16,17]. Typically, a layer of TiN was applied to a bulk substrate by sputter deposition prior to the CNT growth to act as a diffusion barrier between the metal catalyst and the substrate material. It was observed that a dense layer of TiN can prevent catalyst deactivation by its diffusion into the substrate during the high-temperature CVD process. Consequently, the CNT yield on the TiN barrier layer was increased compared to the results on pristine substrate materials, which was previously shown for silicon [18,19,20], tantalum [14], stainless steel [21,22], and copper substrates [12,23]. The observed enhancement in the FE performance is potentially attributed to a combined effect of the modified CNT yield and the electrical conductivity of the TiN layer [13,14].

This work explores the effect of a 10 nm thin TiN film—generated by plasma-enhanced atomic layer deposition (PEALD)—on the FE properties of CNTs that were grown on different planar and three-dimensional (3D) substrate materials. The CNT growth was carried out by CVD using a combination of the botanical hydrocarbon camphor together with ferrocene as the iron catalyst source. Camphor is frequently used as a precursor for CNT growth because it is an environment-friendly plant product that is cheap, user-friendly as it is non-toxic, and it allows for CNT growth with a very high efficiency [24,25,26,27]. We first compared the structural appearance of the CNTs grown on pristine and on TiN-coated bulk substrates. The results indicate that the PEALD TiN film had no obvious influence on the morphology of the CNTs in contrast to reports from previous studies. Secondly, the influence of the TiN layer on the FE properties of the CNTs was examined for the different substrate types. This part of the study is based on a previous publication that exclusively examined the FE from CNTs on planar substrates [28]. Herein, further investigation of the CNTs’ morphology was carried out, the reproducibility of the FE properties was demonstrated, and the effect of the electrode gap width in our FE setup on the measured I–V curves was analyzed. Thirdly, the CNT growth process was transferred to TiN-coated, 3D-printed polymeric microstructures that were fabricated by direct laser writing based on two-photon polymerization [29,30]. The change to 3D-printed substrates was facilitated by the high conformality of the TiN coating that can be achieved by ALD. [31]. In contrast to the directional coating usually formed by physical vapor deposition techniques, such as sputtering or metal evaporation, shadowing effects are avoided and a homogenous deposition can be realized [32]. A slight enhancement in the FE performance was apparent for the transfer from planar to 3D substrates, possibly caused by the additional geometrical field enhancement factor generated by the microprinted structures. Note that field emitters on 3D-shaped substrates have the capability to achieve higher FE current densities than their planar counterparts because of the effective increase in the CNT density without the enlargement of the cathode’s footprint, while avoiding the intensification of the electrostatic screening effect among neighboring emitters [9,33]. Therefore, the results of our study could pave the way for the future fabrication of highly efficient CNT field emitters on various 3D scaffold geometries.

## 2. Materials and Methods

### 2.1. Sample Preparation

The following three substrate types were used in pristine condition as well as with an additional TiN ALD coating for the subsequent CNT growth: n-type silicon (Si, Siegert Wafer, Aachen, Germany), 100 nm silicon nitride (SiN) on bulk Si (Si-Mat, Kaufering, Germany), and 100 nm thick, free-standing SiN nanomembranes (NMs, 2 × 2 mm^2^, Silson Ltd., Southam, UK). The pristine Si pieces had a resistivity of 5–10 Ω∙cm (given by the supplier), whereas the SiN bulk and NM substrates provided an electrically insulating surface. The 10 nm thick TiN films were deposited in a PEALD system (GEMStar XT-DP^TM^, Arradiance, Littleton, MA, USA) in continuous flow mode at a temperature of 250 °C using tetrakis(dimethylamino)titanium(IV) (TDMAT, Strem Chemicals, Newburyport, MA, USA) as the titanium source in combination with a hydrogen/nitrogen plasma. The detailed description of the TiN cycle can be found in a previous publication [34].

The 3D-printed structure was comprised of truncated cones with a height of 150 µm, a base diameter of 120 µm, and a tip width of 4 µm, which were arranged in an array of 9 × 9 cones with the center-to-center distance set to 250 µm. The cone array was generated by direct laser writing based on two-photon polymerization of the commercial resin IP-Q on a silanized Si substrate using the Photonic Professional GT2 (Nanoscribe, Eggenstein-Leopoldshafen, Germany) with a laser power and a scan speed of 96% and 30,000 µm/s, respectively. Subsequently, a 20 nm aluminum oxide (Al_2_O_3_) film was deposited by thermal ALD to protect the polymeric cone array from destruction through plasma etching during the following deposition of 10 nm TiN by PEALD, which used a reduced temperature of 200 °C. The Al_2_O_3_ film was generated in an in-house-built ALD setup from the precursors trimethylaluminum (TMA, Sigma Aldrich, St. Louis, MO, USA) and water at a deposition temperature of 150 °C using nitrogen as the carrier gas (~42 sccm). Both precursors were kept at room temperature and applied alternately in a sequence of pulse (0.05 s), exposure (30 s), and purge (90 s for TMA, 60 s for water). The thickness of the ALD films was measured on Si substrates by spectroscopic ellipsometry (SENpro, Sentech Instruments GmbH, Berlin, Germany).

The CNTs were grown by CVD in a three-zone tube furnace (OTF-1200X-III-UL, MTI Corporation, Richmond, CA, USA) using 500 mg camphor (96% purity, Sigma Aldrich) as the carbon source in combination with 25 mg ferrocene (99% purity, Alfa Aesar, Haverhill, MA, USA) providing the iron catalyst. Both precursors were positioned next to each other in the same ceramic boat without mixing the powders. The precursors were heated to 165 °C in the left zone, while the central zone was set to 300 °C, and a single substrate was exposed to the growth temperature of 900 °C in the right heating zone. These temperatures were applied for the growth duration of 1 h and forming gas (100 sccm, 5% hydrogen/95% argon) was passed through the quartz tube as carrier gas. The CNTs were observed as a black film covering the substrate, the entire ceramic boat, and the inside of the quartz tube in the beginning of the third heating zone of the furnace. Between successive growth processes, the furnace was annealed at 900 °C applying first oxygen (33 sccm) for 2 h and subsequently forming gas (100 sccm) for 10 h, to remove precursor and CNT residues from the inner walls of the quartz tube and the ceramic boat. Note, this process was used in a previous publication exclusively for CNT growth on planar substrates [28].

### 2.2. Structural and Electrical Characterization Methods

The morphology of the CNTs on the different substrate types was analyzed by scanning electron microscopy (SEM) using a Crossbeam 550 system (Zeiss, Oberkochen, Germany). Additionally, energy-dispersive X-ray spectroscopy (EDX) was applied with a voltage of 10 kV in the same system. The CNT dimensions were determined in the SEM images using ImageJ 1.53t [35].

The FE properties of the CNTs were measured in an in-house-built FE setup with a base pressure below 2 × 10^−5^ Pa. For each sample in the triode-type setup, the electric field between the CNT emitter and the nickel grid electrode (59 lpc, 73% transmission, Precision Eforming, Cortland, NY, USA) was increased stepwise, while the current was measured at the anode with a Picoammeter (6485, Keithley, Cleveland, OH, USA). A laser-cut polytetrafluoroethylene (PTFE) sheet between CNTs and grid was used to provide electrical insulation, to set the emitter tip-to-grid distance to 250 µm, and to define the macroscopic emission area to 0.36 cm^2^. One side of the PTEF sheet was coated by 10 nm of titanium as an adhesion layer followed by 50 nm of gold as the electrical contact using electron beam evaporation. The PTFE sheet was placed in the FE assembly with the metal-coated side facing the CNT sample to provide the electrical contact from the CNTs to a separate electrode ring. For a reproducible assembly, a torque wrench was used to fix the stack of the CNT sample, PTFE sheet, and grid. Furthermore, varying the emitter tip-to-grid distance between 250 and 1500 µm for a selected planar sample allowed for investigating the effect of the electrode separation on the I-V curves. Each sample was kept in a high vacuum for 72 h before executing the initial FE measurement to allow for proper outgassing of the setup components. This procedure lowers the risk of an electrical breakdown by the ionization of residual gas molecules between the electrodes. For the same reason, the maximum allowed emission current was stepwise increased from 5 pA to 5 µA for a series of measurements for every new sample.

## 3. Results and Discussion

### 3.1. CNT Morphology

A highly entangled network of randomly oriented CNTs covered the surface of the pristine as well as of the TiN-coated substrates after the CVD growth process, as shown in Figure 1. For each substrate type, a higher magnification SEM image is presented to provide an impression of the detailed CNT morphology. Furthermore, an overview of the CNT layer is given as an indication for potential effects on the density of the CNT network. No distinct differences are obvious for the structural appearance of the CNTs that were generated on the pristine bulk and free-standing NM substrates (Figure 1a–c). Similarly, the direct comparison to the CNTs on the TiN-coated substrates did not reveal any substantial changes in terms of morphology or network density (Figure 1d–f). Hence, no significant impact of the substrate type, whether pristine or TiN-coated, on the morphology of the CNT layer was observed. Consequently, the herein used CNT growth process seems to be largely unaffected by the examined planar substrate types, potentially allowing universal application on both conductive and electrically insulating bulk and membrane substrates.

The morphology of the CNTs was further investigated by measuring their diameter, as summarized in Figure 2. The mean diameter varies between 37 and 42 nm for the CNTs grown on pristine substrates (Figure 2a). A slightly larger range of 37 up to 48 nm was found for the CNT diameters on the TiN-coated substrates (Figure 2b). However, the extracted values vary within their standard deviations and thus, the CNT diameter appears to be independent of the substrate type. An intentionally generated gap in the CNT layer (Figure 2c) reveals a height of roughly 10 µm for the dense network of entangled CNTs. An EDX line scan of this area is presented in Appendix A of the Appendix A, indicating the transition from the CNT layer to the TiN-coated SiN substrate.

Previous studies employing camphor in combination with ferrocene as precursors for CVD also yielded densely packed CNT films with a similar appearance as observed for our growth parameters. Moreover, comparable diameters between 20 and 60 nm were reported for multi-wall carbon nanotubes [24,25,36]. Thus, we assume that multi-wall CNTs were produced in this study based on the observed tube diameters. The relatively broad diameter distribution, which was also observed in our study, is likely related to the in situ generation of iron clusters by the pyrolysis of the ferrocene vapor, which is known as the floating catalyst method. The random formation of iron clusters of different sizes can lead to variation in the CNT diameter because the diameter correlates with the catalyst particle dimensions [27,37,38]. Furthermore, it was proposed that the tendency to form curved, convoluted tubes rather than straight shapes could be related to the hexagonal and pentagonal carbon rings provided by the camphor precursor, which act as the building blocks for the CNTs [25]. However, other studies also reported on the generation of vertically aligned CNTs by CVD using a camphor–ferrocene precursor mixture [39,40,41,42,43]. A significant difference to our three-zone growth oven is that two separate furnaces were used instead. This configuration allowed the heating of the furnace with the substrate to its set temperature first and subsequently, to start the vaporization of the precursors in the other furnace, which prevented heat transfer from one zone to the other, and thus unwanted precursor evaporation during the heating ramp [41,42,43]. Other researchers advanced this practice by shifting the quartz tube, and thereby the precursors, out of the heating zone until the furnace containing the substrate reached its set temperature [39]. Such a procedure is not possible in our three-zone tube furnace as the heating of one zone inevitably also affects the temperatures in the other zones, and it is not possible to shift the tube to any side. Furthermore, Porro et al., who used a separate flask to vaporize the precursor mixture, specified that vertically aligned CNTs were only formed for a certain combination of growth temperature and ferrocene-to-camphor ratio [40].

Herein, we demonstrated that the synthesis of dense CNT layers from the ferrocene–camphor precursor combination is possible in a three-zone tube furnace, but it is likely that the available growth oven configuration prevented the formation of vertically aligned CNTs. The precursor vaporization may have begun before the intended set temperatures in the three zones were reached. Possibly, the nucleation of CNT structures already happened in the gas phase before reaching the optimum growth temperature, leading to the observed CNT deposition on the entire inner surface of the quartz tube in the beginning of the third heating zone, and to the formation of randomly oriented CNTs regardless of the substrate type. Note, this study focused on the effect of the substrate’s TiN coating on the CNTs and their FE properties. Therefore, the optimization of the CNT growth parameters was not pursued further.

A key result of the morphology investigation is that the CNT structures seem to be unaffected by the different substrate types as well as by the TiN coating. In previous studies, TiN films were explicitly used as barrier coatings to prevent the deactivation of the iron catalyst by physical or chemical interactions with the bulk substrate. In contrast to our observations, the TiN films had substantial effects on the yield, growth rate, or the appearance of the CNTs in the other research work [12,18,23,44]. For instance, curved, sparsely distributed CNTs were generated on a SiN-coated copper substrate, whereas a dense mat of aligned CNTs formed on the same substrate type with a TiN barrier layer [23]. We propose that the TiN films in our study had no considerable impact on the CNT density and morphology, which could be related to the potential gas-phase nucleation of the CNTs in our three-zone tube furnace. The interaction of the precursor components with the substrate may have occurred primarily after CNT formation had already started in the gas phase, which could explain the not apparent influence of the tested substrate types.

The iron clusters from the ferrocene precursor appear to be randomly scattered over the CNTs, as shown in the SEM image in Figure 3a. As expected, the EDX spectrum of this area reveals a high carbon abundance and the signal from the Si substrate is clearly detected. However, only a low iron content was found for the EDX scan of the entire area. For comparison, an EDX point scan was performed at the position of the bright particle that is marked by the red arrow in the inset of Figure 3b (spectrum plotted in red). The larger iron content at this position suggests that the bright particle is in fact an iron cluster. Similar to the results in (a), carbon generated the strongest signal and the substrate components, namely titanium and silicon, were detected. Moreover, the oxygen content also increased at the indicated position, which may imply an enhanced amount of oxygen incorporated in the iron cluster. In contrast to the EDX spectrum measured at the location of the bright particle, lower iron and oxygen abundances were found for the CNT structure that is marked by the green arrow in Figure 3b (spectrum plotted in green) similar to the results of the pristine sample.

Although a high carbon content (>70%) in the EDX spectra is expected from previous studies using the ferrocene–camphor mixture, we found an iron content of about 3.2% (weight percent from Figure 3a), which is larger than reported by other researchers (<0.4%) [24,25]. This observation is likely related to the higher ferrocene–camphor weight ratio of 5%, which was used in the current study. Kumar et al. proposed that the optimal ratio is 1%, as too high ferrocene concentrations enhance the deposition of metal particles on the CNTs [24,25]. Therefore, choosing a smaller ferrocene–camphor weight ratio in the future could reduce the metal contamination on our CNT structures. Additionally, we identified the bright particles randomly scattered over the CNTs as iron clusters (Figure 3b). Since no confinement of the catalyst to a specific site of the CNTs was observed, a classical tip or base growth regime was not recognized here [39]. Moreover, an increased amount of oxygen was noticed at the location of the iron cluster. On the one hand, the iron particle was possibly oxidized by the oxygen liberated from the camphor precursor (C_10_H_16_O), which was also proposed in other studies [25,38,45,46]. On the other hand, the iron oxidation could have occurred after the CNT growth process by the inevitable exposure to ambient conditions. Post-deposition annealing was proposed to effectively purify as-grown CNT structures, reduce structural defects, and enhance their graphitization [43,47]. However, for the removal of iron oxide particles, additional treatments in a hydrogen environment or with liquid acids would be needed [47]. These findings have to be considered for the interpretation of the FE data because the iron cluster decoration may influence the electronic surface properties of the CNTs [48].

### 3.2. Field Emission from CNTs on Planar Substrates

The FE measurements from the CNTs on the pristine (hollow symbols) and TiN-coated substrates (filled symbols) are summarized in Figure 4a–c. The mean turn-on field for a macroscopic emission current density of 10 µA/cm^2^, which equals a measured current of 3.6 µA in our FE setup configuration, is presented for each substrate type in Figure 4d. In all three cases, the turn-on field was significantly reduced when an additional TiN coating of the substrate was performed prior to the CNT growth. However, the absolute values vary with the substrate type. The strongest turn-on field reduction of about 59% was found for the SiN substrate followed by a decrease of 30% for the n-doped Si and of 17% for the SiN NM. Note that the macroscopic electric field was defined as the ratio of the applied voltage to the electrode distance of 250 µm.

The general turn-on field reduction is likely related to the good electrical conductivity of the TiN film, which allowed for an efficient electron supply across the substrate towards the CNTs and, possibly, generated a lower CNT–substrate contact resistance [9,10,49]. Furthermore, the highest turn-on field of 2.9 V/µm was found for the pristine, electrically insulating SiN, which emphasizes the influence of the substrate’s electrical properties on the CNTs’ FE performance. As expected, the turn-on field for the pristine Si substrate (1.5 V/µm) is smaller than for the SiN because of its lower resistivity. However, the turn-on field for the pristine SiN NM (1.6 V/µm) is smaller compared to the one found for the SiN bulk support. This observation could be related to the potential deformation of the CNT-covered SiN NM in the electrostatic field during the FE measurement. Mechanical displacement of the membrane towards the grid electrode due to electrostatic force would reduce the electrode distance as a function of the applied voltage. Consequently, the electric field on the surface of the emitter would increase, which leads to the onset of FE at lower applied voltages [50]. Previously, we found that the applied voltage needed to allow for FE from ZnO wires on SiN decreases for the change from a bulk to a flexible NM substrate, which agrees well with the observations in the current study [51].

The turn-on fields for the CNTs on the TiN-coated substrates vary within a narrow range of 1–1.4 V/µm. The lowest turn-on field of about 1 V/µm was found for the TiN-coated Si substrate and a slightly larger value of 1.2 V/µm was extracted for the TiN-covered bulk SiN sample, which indicates a certain influence of the material underneath the TiN film on the electrical properties of the substrate’s surface. The highest turn-on field for the TiN-coated SiN NM (1.4 V/µm) can possibly be attributed to the modification of the mechanical properties of the NM by the additional TiN layer, which may have caused a different displacement behavior of the membrane in the electrostatic field. Overall, the range of turn-on fields identified in this work is similar to previously reported values of 1–2.6 V/µm for CNTs grown by CVD with a camphor–ferrocene mixture on Si substrates [41,48]. A threshold field could not be extracted because the emission current density of 10 mA/cm^2^ was not reached in this study. The main reason for this observation may be the limited transmission of the grid electrode (about 73%) in our triode-type FE measurement setup leading to charge accumulation in the emitter–grid gap, possibly shielding the CNT surface from the electric field [51]. Furthermore, a lower ferrocene content could be tested in the future to decrease the threshold field [48] or nitrogen doping to reduce the CNTs’ effective work function [52].

In Figure 4a–c, a distinct change in the slope of the FE data becomes obvious in the microampere range. Such an emission current saturation is often observed for the FE from CNTs and may be attributed to several reasons, such as the effect of adsorbates on the emitters’ surface electronic properties [53], formation of vacuum space charge in the electrode gap [51,54], and electrical resistance of the substrate–CNT interface [55,56]. Therefore, only the initial linear increase in each I-V curve was considered for the extraction of the apparent field enhancement factor (FEF) using the Murphy–Good (MG) plot, which is shown as an example in Appendix A of the SI [57]. Table 1 gives a summary of the turn-on fields and apparent FEFs for the CNTs on the planar substrates. A work function of 5.0 eV was assumed for the analysis of the FE data from the carbon-based emitters as it was done in previous publications [14,41,49,58].

The apparent FEFs for the CNTs on the pristine substrates vary in a wide range between 1169 and 3591, which suggests that these values were not solely determined by the geometrical shape of the emitters. This hypothesis is supported by the not obvious change in the CNT morphology with the substrate type (see Section 3.1). Herein, the apparent FEFs may rather reflect the influence of the substrate’s electrical conductivity on the CNTs’ FE properties since the values increase for the transition from the pristine to the TiN-coated substrate in all three cases. Additionally, the so-called orthodoxy test was applied to the FE data in the MG plot [59]. The FE measurements from the CNTs on the TiN-coated Si bulk and on the TiN-coated SiN NM substrate passed the orthodoxy test, whereas the results for the other substrate types are inconclusive, as summarized in Appendix A of the SI. These findings indicate that the electron emission was possibly affected by other features of the measurement system, including the sample itself as the major component, that are not part of the conventional theory used for the FE data analysis [59,60]. On the one hand, a possible explanation for the deviations from the theoretical description could be a higher contact resistance for the CNTs on the pristine substrates. On the other hand, the surface electronic properties of the CNTs may have been affected by the iron clusters that were found on the structures (see Figure 3) [48].

The turn-on field for a sample comprised of a CNT layer on a Si substrate as a function of the emitter tip-to-electrode distance (sample A, plotted in black) is shown in Figure 5. It is obvious that the turn-on field decreases with the increasing electrode gap width approaching an asymptotic value for distances larger than 500 µm. This observation agrees well with previous studies that reported a similar behavior for the electric field needed for FE, which was attributed to an effective reduction in the field enhancement factor for smaller electrode gaps [61,62]. These results suggest the use of an electrode gap distance of at least 500 µm for future FE measurements to mitigate this effect. Moreover, the turn-on field that was extracted for the CNTs on the pristine Si substrate in Figure 4a was added to Figure 5 as ‘CNTs on Si (sample B)’ (plotted in blue). The turn-on field of sample B lays within the standard deviation of the turn-on field measured for the other sample of the same type, namely ‘CNTs on Si (sample A)’, which demonstrates the reproducibility of our CNT growth process and of the FE properties.

### 3.3. Field Emission from CNTs on 3D-Printed Substrates

To further investigate the influence of microstructured complex substrates on the FE properties of CVD-grown CNTs, 3D-printed cones were used as the underlying substrate. The array of polymeric cones was first covered by 20 nm Al_2_O_3_ using thermal ALD as a protection against potential plasma etching during the subsequent PEALD of 10 nm TiN. The ALD-covered cone structures remained intact, as shown in Figure 6a. After the following CVD growth process, a dense layer of CNTs was wrapped around the cones (Figure 6b). The measured current from the CNTs on the 3D structures is shown as a function of the macroscopic electric field for an electrode distance of 500 µm (plotted in black) and of 250 µm (gray) in Figure 6c. As expected from the results shown in Figure 5, the turn-on field increased from (0.98 ± 0.04) V/µm to (1.13 ± 0.02) V/µm when the electrode distance was reduced. Therefore, the FE results from the CNTs on the planar TiN/SiN sample will only be compared to the FE data from the CNTs on the TiN/Al_2_O_3_/3D substrate measured with the 250 µm electrode distance. The turn-on field slightly decreased for the transition from the planar ((1.20 ± 0.03) V/µm) to the 3D-printed substrate ((1.13 ± 0.02) V/µm) and the apparent FEF extracted from the MG plot of the FE data increased from 3681 ± 386 to 4165 ± 742. A stability test for the FE from the CNTs on the 3D-printed structures was performed for 20 h and is presented in Appendix A of the SI.

Most cones of the array remained intact during the sequence of ALD coating, CVD growth, and subsequent FE measurements. However, a few structures were completely removed or detached from the Si substrate, as shown in Appendix A of the SI. It is likely that the polymeric material was destroyed during the high-temperature CVD step, but the conical shape was often preserved by the CNTs grown on the ALD layers. Possibly, thicker ALD films could help to maintain the structural integrity of the 3D-printed substrates during the CNT growth process in future. Furthermore, the recently introduced 3D-printed fused silica glass could be explored as a scaffold material for the field emitter growth because of its high thermal and chemical stability [63]. As shown in Appendix A of the SI, a dense CNT layer was generated on such a 3D-printed glass microstructure. Since the glass detached from the Si substrate during the CVD process, probably caused by a mismatch of the thermal expansion coefficients, no FE measurements could be performed for this sample type. In future experiments, the adhesion of the glass structures needs to be improved for the high-temperature CVD step, possibly by adjusting the dose for printing the structure’s base or by an additional substructure at the interface between the glass and substrate.

The turn-on field was reduced by about 6% for the transition from the planar to the 3D-printed substrate, while the apparent FEF increased slightly. This observation was probably caused by the additional geometrical FEF generated by the conical microstructure underneath the CNT layer. A theoretical FEF of 7.4 was calculated for the 150 µm tall structure using the “hemi-ellipsoid on a plane” model, which may be applied when the structure’s apex width (4 µm) is much smaller than the base diameter (120 µm) [64]. For a small protrusion that sits on top of a much larger structure, a multiplicative behavior of their individual FEFs is expected, which is known as Schottky’s conjecture [65]. Therefore, a seven times larger apparent FEF than found for the CNTs on the planar substrate would have been expected, but the values extracted from the experimental FE data vary within their standard deviations. The deviation from the theoretical prediction may be related to the difference in effective emission area. On the planar substrate, a 0.36 cm^2^ CNT-covered area was exposed to the electric field, whereas for the 3D substrate, the CNTs on the micrometer-wide apex area of the cones may have dominated the FE. Further investigation of those effects is needed to explore ALD-covered, tailored, and 3D-printed scaffold materials for the fabrication of efficient field emitters.

## 4. Conclusions

CNTs were directly grown by CVD on different pristine and TiN-coated substrates. No distinct effect on the CNTs’ structural appearance was observed, but their FE properties were considerably enhanced by the additional TiN coating identified as a turn-on field reduction of up to 59%. These results were also applied to free-standing, electrically insulating NM substrates, revealing an improved FE performance, which may allow their future use for FE-based sensor applications, such as in a NM detector for mass spectrometry of high-mass proteins [50,66]. Furthermore, 3D-printed microstructures were investigated as a scaffold material for the CNT growth as a proof-of-concept study. Only a small improvement in the FE properties was found for the transition from the planar to the 3D-printed substrate, which may be related to the reduced effective emission area. Hence, further investigation of the influence of the 3D-printed structure’s geometrical shape on the FE from the CNTs is needed. However, we herein prove that the combination of 3D printing by direct laser writing and ALD coating provides the opportunity to generate scaffold structures of arbitrary shape resistant to the subsequent high-temperature CNT growth and supplying an electrically conductive surface for the FE application. Such a fabrication route with state-of-the-art methods, i.e., 3D printing and ALD, could assist the generation of tailor-made FE electron sources in the future.

## Figures and Tables

**Figure 1 nanomaterials-14-00781-f001:**
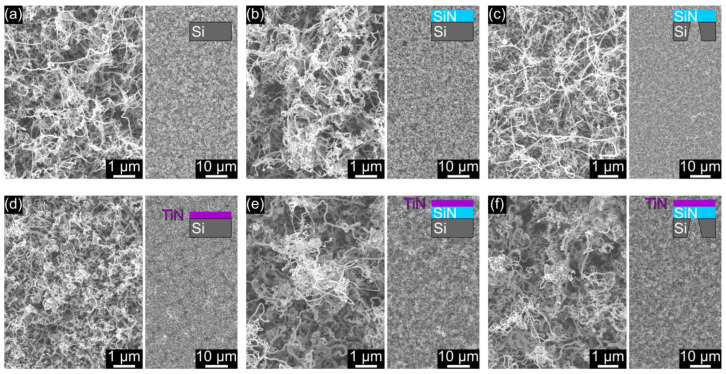
In the top row, SEM images are shown of CNTs grown on the following pristine substrates: (**a**) Si, (**b**) SiN, and (**c**) a free-standing SiN NM. In the bottom row (**d**–**f**), CNTs are presented on the same substrate types that were coated with an additional TiN film before the CNT growth. The sketches in the insets indicate the layer sequence of each substrate type.

**Figure 2 nanomaterials-14-00781-f002:**
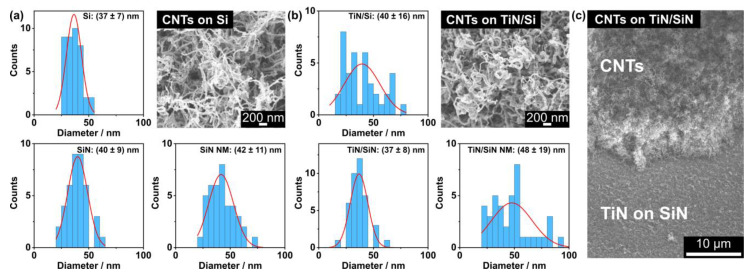
Diameter distributions for the CNTs grown on (**a**) pristine and (**b**) TiN-coated substrates. SEM images of CNTs on a pristine Si and on a TiN-coated Si substrate are presented as examples for the magnification that was used to measure the CNT diameters. (**c**) SEM image of a cross section through the CNT layer on a TiN/SiN substrate, which was generated by using tweezers.

**Figure 3 nanomaterials-14-00781-f003:**
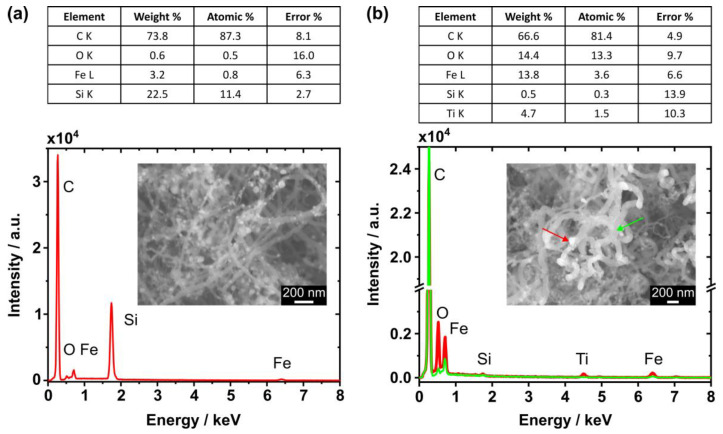
(**a**) EDX spectrum of CNTs on a pristine Si substrate obtained from the area shown in the inset. (**b**) EDX spectra of CNTs on a TiN-coated SiN NM measured at the locations indicated by the arrows in the inset SEM image. The locally increased iron abundance in the spectrum (plotted in red) suggests that the bright particles are iron clusters (red arrow). For comparison, a lower iron abundance is observed in the green spectrum measured at the CNT structure (green arrow). The tables present the quantification results of the spectra plotted in red.

**Figure 4 nanomaterials-14-00781-f004:**
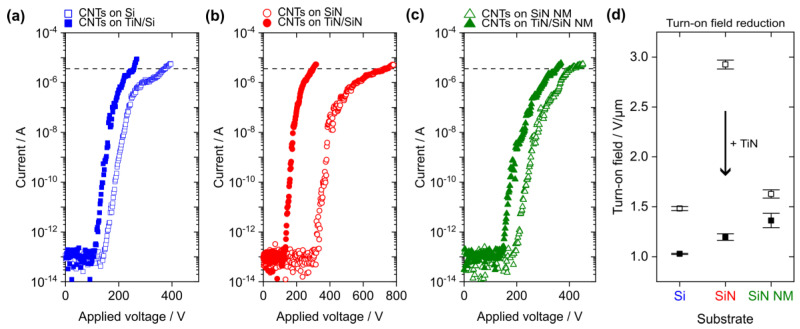
Measured current as a function of the applied voltage for the CNTs grown on the following pristine (hollow symbols) and TiN-coated substrates (filled symbols): (**a**) bulk Si, (**b**) SiN on bulk Si, and (**c**) SiN NM. The mean of three subsequent I–V curves is presented for each substrate type and the horizontal dashed lines indicate the current threshold (3.6 µA) used for the turn-on field extraction. (**d**) Reduction in the mean turn-on field by TiN coating of the substrate. The error bars present the turn-on field variation in the three subsequent FE measurements for each substrate type.

**Figure 5 nanomaterials-14-00781-f005:**
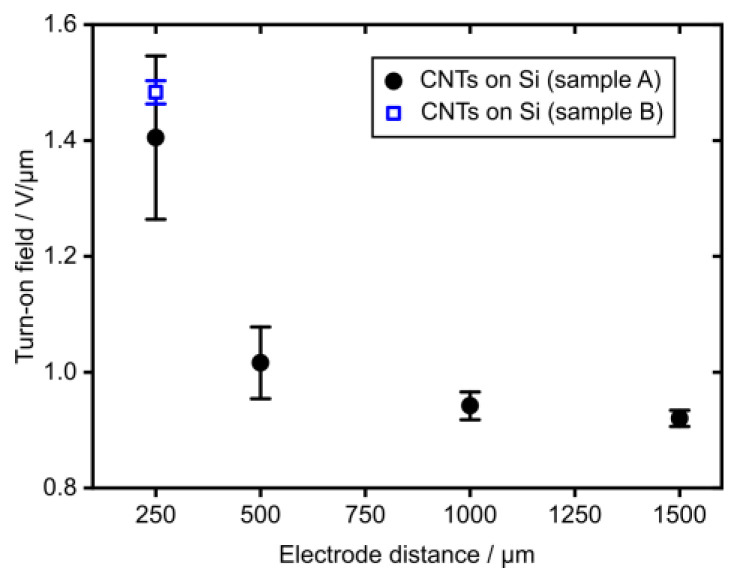
Turn-on field reduction with increasing electrode distance (sample A) and turn-on field for 250 µm (sample B). Both samples comprised of CNTs on a pristine Si substrate and the electrode distance was defined by PTFE spacers of different thicknesses.

**Figure 6 nanomaterials-14-00781-f006:**
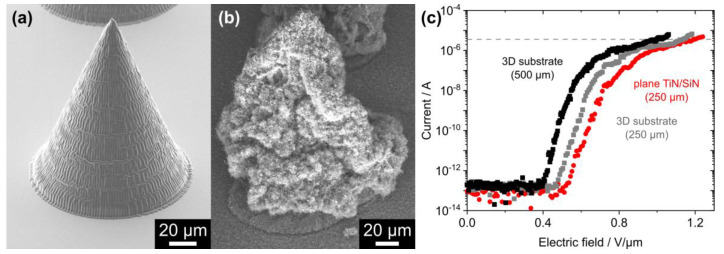
(**a**) The 3D-printed polymeric cone covered with 10 nm TiN on top of 20 nm Al_2_O_3_ before CNT growth. (**b**) CNT layer on an ALD-coated polymeric cone. (**c**) FE measurements from an array of CNT-covered polymeric cones for an electrode distance of 500 µm (plotted in black) and of 250 µm (gray). For comparison, the FE data from the CNTs on a planar TiN/SiN sample is shown (250 µm, red). Each curve represents the mean of three subsequent FE measurements and the horizontal dashed line marks the current threshold (3.6 µA) for extraction of the turn-on filed.

**Table 1 nanomaterials-14-00781-t001:** Mean turn-on field and apparent field enhancement factor (FEF) for the CNTs on the pristine and TiN-coated planar substrates.

Substrate	Turn-On Field/V/µm	Apparent FEF
Si	1.48 ± 0.02	3591 ± 552
TiN on Si	1.03 ± 0.01	4974 ± 649
SiN	2.93 ± 0.04	1169 ± 49
TiN on SiN	1.20 ± 0.03	3681 ± 386
SiN NM	1.63 ± 0.04	2404 ± 272
TiN on SiN NM	1.36 ± 0.07	4042 ± 865

## Data Availability

The data supporting the conclusions of this article will be made available by the authors upon reasonable request.

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
