# Peer review of "Field Emission from Carbon Nanotubes on Titanium Nitride-Coated Planar and 3D-Printed Substrates"

_nanomaterials, 2024, doi:10.3390/nano14090781_

Round 1
Reviewer 1 Report
Comments and Suggestions for Authors
The manuscript presents results of experimental investigations on production of carbon nanotube layers and characterization field emission properties of the cathode with different substrate compositions and topology features.
Similar imvestigations in the both aspects (carbon nanotube growth and field emission characterizations) were reported in details before. And. thus, the present investigation posses mostly 'technical' interest rather than 'academic'.
Moreover some important details in the research design and presentation should be clarified additionaly.
1. It is necessary clearly state about type of used carbon nanotubes. Is it single wall or multiwall nanotubes? Experimental data or suitable references on this matter should be provided. Also level of structural defects in the tubes is importnat issue for interpretation of obtained results - corresponding information should be added.
2. As grown layer of carbon nanotubes is composed of randomly oriented wires. But under action of applied voltage these wires (CNT) should be pulled out by the field and oriented along the field direction. Length and distance between the standing wires will determine field enhancement factor. Thus some additional information or modelings are necessery to estimate possible effect of these factors on field emission characteristics and compare this effect with effect of back-contact resistance which is assumed in the work as the main reason for FE properties variations.
3. PTFE layer is used as a spacer in the FE measurements. It is necessary explain how this spacer has been attached to the sample and tothe grid: is it somehow pressed or not? Criticaly importnat issue is surface conductivity of the spacer material. Taking into account huge electric field applied this conductivity, which always exists and cannot be removed completely, might affect on results of measurements of the current-voltage dependencies. The authors should explain how they taken it into account to eliminate possible problems in measurements and interpretations.
Reviewer 2 Report
Comments and Suggestions for Authors
This paper gives an intense study on the influence of the substrate material to the field emission performance of CNT electron sources, which will be surely beneficial to future research. I recommend the paper be accepted.
Please consider comments below:
11. The authors claim TiN as “a suitable material for the CVD growth of CNT field emitters with a high melting point that has a low electrical resistivity”. What if replace TiN with other conductive materials, such as gold or tungsten, Titanium?
22. The emission current and emission current density of CNTs in this paper seem not good enough compared with previous work (some of them are listed in Vacuum 198 (2022) 110900). Why? How to improve?
33. CNT field emission has been investigated for many years, but it is still not rarely used. One of the most important factors is its poor lifetime and stability. What’s the stability of the CNT electron sources in vacuum, or how long can they work for?
44. The 3D-printed polymeric cones seem not uniform and cannot bring great improvement on the emission performance including the turn-on field or emission current density. So, what’s the significance of such a structure?
Reviewer 3 Report
Comments and Suggestions for Authors
This is an interesting paper with clear and meticulously description. There are a few words which should be changed to conform to standard English idiom and there is some minor clarification needed in lines 368-370 where the difference between samples A and B should be explained in the text as well as in the legend of Fig.5.
Comments on the Quality of English Language
There is the occasional word which should be changed to conform to standard English:
line 36 "humongous" is not an orthodox adjective for a scientific paper. It could be replaced by "very large".
lines 193 and 336 "Exemplarily" is not a word used in standard English. I suggest "As an example" or "for example" instead.
